# Evaluating How Mental Health Changed in Australia through the COVID-19 Pandemic: Findings from the ‘*Taking the Pulse of the Nation*’ (TTPN) Survey

**DOI:** 10.3390/ijerph19010558

**Published:** 2022-01-04

**Authors:** Ferdi Botha, Peter Butterworth, Roger Wilkins

**Affiliations:** 1Melbourne Institute: Applied Economic & Social Research, University of Melbourne, Melbourne 3052, Australia; peter.butterworth@anu.edu.au (P.B.); r.wilkins@unimelb.edu.au (R.W.); 2ARC Centre of Excellence for Children and Families over the Life Course, Indooroopilly 4068, Australia; 3Research School of Population Health, Australian National University, Canberra 2601, Australia; 4Institute of Labor Economics (IZA), 53111 Bonn, Germany

**Keywords:** psychological distress, COVID-19, mental health, measurement

## Abstract

The COVID-19 pandemic has had a significant impact on mental health at the level of the population. The current study adds to the evidence base by examining how the prevalence of psychological distress changed in Australia during the pandemic. The study also assesses the psychometric properties of a new single-item measure of mental distress included in a survey program conducted regularly throughout the pandemic. Data are from 1158 respondents in wave 13 (early July 2020) of the nationally representative Taking the Pulse of the Nation (TTPN) Survey. The questionnaire included the six-item Kessler Psychological Distress Scale (K6) and a new single-item measure of distress. Results show a significant increase in the prevalence of psychological distress in Australia, from 6.3% pre-pandemic to 17.7% in early July 2020 (unadjusted odds ratio = 3.19; 95% CI (confidence interval) = 2.51 to 4.05). The new single-item measure of distress is highly correlated with the K6. This study provides a snapshot at one point in time about how mental health worsened in Australia during the COVID-19 pandemic. However, by demonstrating the accuracy of the new single-item measure of distress, this analysis also provides a basis for further research examining the trajectories and correlates of distress in Australia across the pandemic.

## 1. Introduction

The COVID-19 pandemic has had a significant impact on mental health at the level of the population, through both the direct effects of COVID-19 (e.g., fear of catching the virus) and the indirect effects associated with the policy responses used to contain the spread of the virus (e.g., lockdowns leading to social isolation and job loss [1,2,3]). In Australia, several studies have investigated mental health during COVID-19 using a variety of different approaches. A number of longitudinal cohort studies underway prior to 2020 have collected data during the pandemic, providing measures from the same individuals before and during the pandemic, e.g., [4,5]. Other studies have recruited participants during the pandemic, via online/social media methods, media advertising, random-telephone calling, or from existing panels established by market research companies to be broadly representative of the population, e.g., [6,7,8].

The current study examines data from the Taking the Pulse of the Nation (TTPN) survey which has tracked the economic, social and personal wellbeing, and attitudes of Australians over the course of the pandemic [9]. The survey was initiated by the Melbourne Institute at the University of Melbourne shortly after the onset of the pandemic and used a repeated cross-section time-series design, drawing on a new representative sample each wave to produce a series of national snapshots. To date, there have been 43 separate waves of data collection (see Section 2.1). A number of core measures, including mental health, have been assessed at each wave to monitor trajectories of change over time. The survey program has also collected additional topical content at each wave, and in early July 2020 included a widely used scale of psychological distress. The aims of this paper are to compare population mental health in Australia at the end of the first lockdown with pre-pandemic levels, and to evaluate the robustness of the ultra-short measure of mental health included in all waves of the survey to ascertain if it can be used to investigate trajectories of mental health in the Australian population across the course of the pandemic. 

### 1.1. Measuring Mental Health 

Scales assessing non-specific psychological distress have an important role in mental health research, surveillance/monitoring and clinical practice [10,11,12]. The Kessler Psychological Distress Scale (the 10-item K10 version and the shorter six-item K6) is one of the most commonly used distress scales. The scale was developed to identify those with serious mental illness [11,13], and is most strongly associated with affective and anxiety disorders [14]. The Kessler scale is commonly used to screen for mental illness in primary care, and to evaluate clinical outcomes [12]. In Australia, the K10 is regularly included in surveys conducted by the Australian Bureau of Statistics [15] and since 2007 has been included biennially in the Household, Income and Labour Dynamics in Australia (HILDA) Survey [16,17]. The widespread use and availability of data using the K6/K10 provides a wealth of normative data. We draw on data from the TTPN survey in the week commencing 30 June 2020, and comparative data from the 2019 wave of the HILDA Survey during 2019, to examine the change in psychological distress in the Australian population and key subgroups during the COVID-19 pandemic.

### 1.2. Ultra-Short Measures

While the K6 is a short scale, there are circumstances in which an even briefer measure of distress may be needed. For example, the cost and time needed to add multi-item scales of mental health or psychological distress to large epidemiological or omnibus panel surveys may be prohibitive, whereas the addition of a single item may be feasible [18]. A very brief measure of mental health may aid screening in primary care, where depression and other mental disorders are common, but appointments are short and time pressured [19,20]. Ultra-short measures of mental health/distress offer a potential way to balance the benefits of regular monitoring of mental health with the need to minimise respondent burden. This may be particularly important when assessing co-morbid distress among people with chronic (physical) conditions such as stroke, cancer or heart disease, or in treatment contexts (such as oncology) to assess psychological responses to treatment and promote clinical discussion [21,22,23]. Single-item measures also have an important role in ecological momentary assessment or monitoring distress via mobile devices, such as via SMS text messages [24,25,26]. 

Although these types of need have driven the development and use of ultra-short mental health measures [19,22,23,27,28,29,30,31,32,33], single-item measures have clear limitations [20,34,35]. For example, the Distress Thermometer, a single item visual analogue measure of distress, is recommended and widely used to screen for distress in oncology patients [36]. However, Hughes and colleagues [21] note that distress is a complex concept, and respondents may not understand a single item asking specifically about distress (rather than symptoms or feelings). Other research investigating a mental health parallel to the widely used Self-Rated Health item (e.g., *In general, would you say your mental health is*:) found only moderate correlation with established mental health scales [34]. However, other studies have found single-item measures tied to specific symptoms (either depression or anxiety) can have adequate sensitivity and specificity when compared to diagnostic criteria and moderate to high levels of accuracy or correspondence with longer scales of depression or anxiety [19,23,27,28,29,30,32].

### 1.3. Assessing Psychological Distress throughout the COVID-19 Pandemic

The need for an ultra-short measure of psychological distress in the TTPN survey was to balance the aim of regularly measuring key aspects of economic, social and personal wellbeing throughout the pandemic while minimising respondent burden during this stressful period. Therefore, the core measures in all content areas, including mental health, were restricted to a single-item measure. The new mental distress item was based on the K6 scale.

Formally, the aims of the current study are: (i) to estimate the prevalence of high psychological distress in the Australian population, and among key subpopulations, during the COVID-19 pandemic using the K6, and to compare these with pre-COVID population estimates; and (ii) to assess the psychometric properties and the accuracy of a single-item measure of mental distress in comparison to the K6.

## 2. Materials and Methods

### 2.1. Participants and Procedure

The data emanate from the Melbourne Institute’s TTPN survey, a repeated cross-sectional survey conducted in Australia since April 2020. The aim of the survey is to monitor key indicators of life in Australia such as mental distress, financial stress, satisfaction with Government policy responses, beliefs about the effect of pandemic, and labour force status. Conducted by marketing research company *OZInfo*, the TTPN survey is based on a mixed sampling frame of phone interviews and online responses, with the same set of questions asked in both forms. Phone responses are recorded by the caller, whereas online respondents complete the responses themselves. Each wave recruits 1200 respondents from all states across Australia, although the number of responses from the Northern Territory are often tiny. The sample is stratified by age, gender, and location to be representative of the Australian population, with weights generated for each wave to better reflect the profile of the Australian population. The ethical aspects of the TTPN survey were approved by the Human Research Ethics Committee at the University of Melbourne, under the project title “Social and economic effects of COVID-19 on the Australian population” (Reference number: 2021-14006-14669-1).

The current analysis draws on wave 13 of the TTPN conducted between 30 June 2020 and 3 July 2020. The wave 13 questionnaire included the six-item Kessler Psychological Distress Scale (K6) [11] (only included in wave 13), in addition to the TTPN single-item mental distress measure (included in every wave). At the time of the survey, Australia was on the cusp of the second wave of widespread COVID-19 infections. Lockdown restrictions associated with the first wave of COVID-19 in Australia had been eased from May 2020. However, new case numbers in the state of Victoria reflecting community transmission began to rise from mid-June 2020. At the time of wave 13 of the survey, a number of local areas around the Victorian capital city of Melbourne had moved into lockdown to limit further community transmission. On 8 July (just beyond the scope of this survey), the restrictions were extended to the entire Melbourne city region [37].

### 2.2. Analytical Sample

Of the 1200 participants in wave 13 of the TTPN survey, 25 respondents (3.2%) had missing data on at least one of the K6 items (see Section 2.3(b)), 17 had missing data on the TTPN mental distress item (see Section 2.3(a)), and 13 had missing data on both K6 and TTPN items. A further eight cases had missing data on the measure of labour force status. This analysis uses complete cases and, therefore, the final sample size is 1150. Survey participants excluded with missing data were similar to those without missing data on all characteristics (gender, labour force status, residential location (metropolitan/rural) except for age. Younger respondents were less likely to complete all items ($\chi_{\left(7\right)}^{2}$ = 44.3, *p* < 0.001).

### 2.3. Measures


(a)K6 scale


The K6 is a well-established measure of distress and commonly used to identify serious mental illness [11]. It provides the reference standard for the current analysis. The K6 comprises six items that ask respondents about feeling: nervous, hopeless, restless or fidgety, so depressed that nothing could cheer them up, that everything was an effort, and worthless. The response options are (1) ‘none of the time’; (2) ‘a little of the time’; (3) ‘some of the time’; (4) ‘most of the time’; or (5) ‘all of the time’. As administered in the TTPN survey, respondents were asked about a 7-day period. The K6 usually asks about the past 30 days, but the scale creators suggest the reference period can be changed to suit study design requirements, and reference to feelings in the previous 7 days would not cause significant issues in scale interpretation [38]. The total K6 scale is calculated by summing the items and produces scores ranging between 6 and 30. We used the established cut-point of scores of 18 or higher to define a binary indicator of high levels of psychological distress and likely serious mental illness [38]. Cronbach alpha of 0.926 (bootstrapped 95% CI (confidence interval) = 0.918 to 0.935) for the K6 showed internal consistency of the scale was excellent, and the average item-rest correlation was 0.77 (range from 0.72 to 0.84).
(b)TTPN mental distress measure

The TTPN single distress item was constructed for inclusion in each wave of the survey and designed to (i) be brief, (ii) have face validity and be readily interpretable, and (iii) be based upon the K6 measures. Thus, respondents were asked “*During the past week, about how often did you feel depressed or anxious?*” with the same response options as for the K6. This item directly assesses feelings of anxiety or depression, which is consistent with the item content of the K6 [12,39]. Similar to the K6, this single item was used as a basis for a binary indicator of psychological distress.

In addition to the measures of distress, the analyses consider age, sex, labour force status (employed, unemployed and looking for work, or not participating in the workforce), and location of residence (metropolitan vs. rural; Victoria vs. rest of Australia) given that these characteristics are available in the data and may be associated with distress.

### 2.4. Comparison Sample

To provide a pre-COVID-19 comparison, we use data from the 2019 wave of the Household Income and Labour Dynamics in Australia (HILDA) Survey. The HILDA Survey is a longitudinal household panel survey that commenced in 2001 and, when weighted, provides a nationally representative estimate of Australian households, though underestimates new migrants and those living in very remote areas of the country. For more details of the HILDA Survey, see [40]. The 2019 survey included the K10 scale of psychological distress in the self-completion questionnaire, though we draw on only the six items that comprise the K6. It is important to note that the K10 included an item asking about feeling “*so sad that nothing could cheer you up?*” whereas the matching item in the K6 refers to feeling “*so depressed that nothing could cheer you up?*”. Others have treated these items as similar in analysis [41]. The other difference to note is that the items in the HILDA Survey referred to the experience of each symptom during the usual past 30-day period. Analysis of the HILDA Survey data was restricted to those are 18 years or older to match the TTPN sample.

### 2.5. Statistical Analysis

We initially present unweighted descriptive statistics for the pre-COVID HILDA and TTPN samples, and compare labour force status in weighted analysis with the expectation that the TTPN survey will demonstrate a significant increase in unemployment associated with the COVID-19 pandemic. We then report results of a series of (weighted) simple logistic regression models, comparing the prevalence of high psychological distress across the two occasions, and stratified by key demographic characteristics to identify whether, and in which social groups, levels of psychological distress had worsened during the pandemic. A final (weighted using the TTPN Survey weights) logistic regression model includes all covariates.

To evaluate the new single-item measure of mental distress, we directly compare a binary estimates of distress (weighted and unweighted) produced by the TTPN item with the standard K6 measure, and examine the correlation and Cohen’s Kappa coefficient. The predictive accuracy of the TTPN item was assessed by receiver operating characteristic (ROC) curve analysis, and examination of Area Under the Curve (AUC) [42]. We report measures of sensitivity (probability that someone identified with distress on the K6 is similarly identified by the TTPN item), specificity (that someone without distress according to the K6 is similarly classified by the TTPN item), positive predictive value (PPV; the probability that those identified as distressed by the TTPN item are identified as distressed by the K6) and negative predictive value (NPV; those not identified as distressed on the TTPN item are not identified as distressed by the K6), as well as the accuracy of classification at each response category on the TTPN item.

The final analyses present a series of random-effects logistic regressions to directly compare the two binary distress measures available for each participant (based on the K6 scale or the TTPN single-item measure). The test of the level-2 indicator evaluates the difference in the estimate of the prevalence of psychological distress derived from the two different measures within the same individuals, while the test of an interaction between the distress measure and each demographic or economic variable (e.g., age, gender, location, and employment status) assesses the consistency of the two measures within the different subpopulations.

## 3. Results

### 3.1. Estimating the Prevalence of Distress in 2019 and 2020

Characteristics of the analysis samples are presented in Table 1. The HILDA Survey sample is much larger than the TTPN sample, but does not include the single-item measure of psychological distress. The TTPN sample comprises a similar number of men and women, and includes representation across all age categories. Comparison of the unweighted sample characteristics from the two different surveys (final column of Table 1) shows there are significant differences in profiles of age, location, labour force status, and mental distress. For example, over 12% of respondents in the TTPN sample are identified as unemployed compared to 3.4% in the 2019 HILDA Survey sample. However, the application of survey weights removed the differences in stable population characteristics. The weighted estimates of gender (*χ*^2^ (1) = 0.01, *p* = 0.9), age group (*χ*^2^ (6) = 1.48, *p* = 0.96) and location (*χ*^2^ (1) = 1.10, *p* = 0.30) were similar in the 2019 and 2020 samples. There was, however, a significant difference in labour force status (*χ*^2^ (df = 2) = 181.70, *p* < 0.001), with estimated unemployment increasing from 2.7% of the population in 2019 to 12.0% in 2020.

The weighted estimates of the prevalence of psychological distress in Australia from the two surveys, and by key characteristics, are presented in Figure 1, together with the results of logistic regression analysis (presenting Odds Ratios and 95% confidence intervals). The overall K6 estimate of high psychological distress increased from 6.3% in the 2019 HILDA Survey data to 17.7% in the 2020 TTPN data, with an odds ratio of 3.19. This odds ratio indicates that the odds of psychological distress were over three times higher in the 2020 survey than prior to COVID-19. The mean K6 scale score increased from 10.2 pre-COVID to 12.6 in 2020, which represents a significant different of 2.5 scale points (95% CI: 1.98–2.93).

A series of separate analysis found there were significant interactions between gender and timing (pre- vs. COVID-19; *χ*^2^ (1) = 10.19, *p* < 0.001), age category and timing (*χ*^2^ (2) = 9.28, *p* < 0.01) and labour force status and timing (*χ*^2^ (2) = 22.82, *p* < 0.001). Figure 1 shows that the increase over time in the reported prevalence of distress for men was much greater than that reported by women (odds ratios of 3.23 vs. 1.70). The increase in the prevalence of distress was inversely related to age: the difference in distress between 2019 and 2020 was most pronounced among younger people, while no change was evident for those aged 65 years and older. Among the labour force status categories, the prevalence of distress only increased among the employed group, with no significant change in distress for those who were unemployed or not participating in the labour force between 2019 and 2020. Considered another way, while people who were unemployed reported much higher levels of distress than those who were employed in 2019, there was no difference in 2020.

Analysis comparing the prevalence of psychological distress over time in the state of Victoria (where new COVID-19 case numbers had begun to increase at the time of the TTPN survey) and the rest of Australia demonstrated no difference (interaction term between time and location: OR = 0.88, 95% CI: 0.54–1.44). A final logistic regression model controlling for all covariates continued to demonstrate a significant effect of timing (pre- vs. COVID-19; OR = 3.03, 95% CI: 2.36–3.90).

### 3.2. Association between Taking the Pulse of the Nation (TTPN) Single Distress Item and the K6

There was a strong positive correlation between the K6 scores and the TTPN distress item (Pearson’s correlation = 0.81; polyserial rho = 0.82). In comparison, the correlation between each item within the K6 and the scale score constructed from the remaining K6 items (item-rest correlation) ranged between 0.72 to 0.84, with a mean of 0.79.

Figure 2 shows that the mean K6 score increases across levels of the TTPN single-item responses. All pairwise comparisons between adjacent categories on the TTPN item were significantly different. The mean increase on the K6 scores between levels of the TTPN item were 3.5 (between *none* to *little*; 95% CI: 2.94–3.99), 4.1 (*little* to *some*; 95% CI: 3.58–4.70), 4.1 (*some* to *most*; 95% CI: 2.25–4.75) and 6.3 (*most* to *all*; 95% CI: 5.18–7.33).

Using the standard cut-point on the K6 for comparison, the ROC curve presented in Figure 3 plots the sensitivity against (1-specificity) for each value of the TTPN distress item. The measure of AUC was 0.93 (95% CI: 0.91–0.95), indicating the TTPN item has a high level of accuracy in predicting the categorical measure of distress derived from the K6.

Table 2 presents the measures of sensitivity, specificity and other measures of the correspondence between the K6 and the TTPN distress item. The optimal cut-point for the TTPN distress item is between the “some of the time” and “most of the time” categories. This cut-point accurately classified 89% of respondents according to the K6 categories and showed adequate sensitivity (71%) and good specificity (93%). The direct comparison of unweighted estimates of distress using the K-6 and TTPN item were 15.3% (95% CI: 13.3–17.5) vs. 16.8% (95% CI: 14.7–19.1), whereas the weighted comparison was 17.7% (95% CI: 14.8–21.1) vs. 18.1% (95% CI: 15.2–21.3). Cohen’s Kappa for these two measures was 0.62, indicating substantial agreement.

Finally, a series of random-effects logistic models tested whether there were differences between the two estimates of distress derived for each individual from the TTPN item and the K6 scale. An initial model including all the covariates described in Table 3 found no significant effect of the measure (OR = 1.07; see column 1), suggesting the estimates of distress based on the TTPN and K6 did not significantly differ. The models presented in Table 3 show no evidence of a significant interaction between the measure (distress based on TTPN or K6) and any of the socio-demographic characteristics, apart from age. The addition of the interaction between the type of distress measure and age category significantly improved model fit over the main effects model ($\chi_{\left(2\right)}^{2}$ = 8.15, *p* = 0.017). Calculating the marginal probabilities showed that the estimated prevalence of distress derived from the two measures (K6 and the TTPN item) did not significantly differ for those aged 18 to 34 years (24.40 vs. 22.3%; difference = –2.12, 95% CI: –6.12–1.88) or those aged 36 to 64 years (14.13 vs. 16.84; difference = 2.71, 95% CI: 4.51–7.33), but that the prevalence estimate derived from the TTPN item was significantly greater than from the K6 for those aged 65 years or older (4.70 vs. 8.59%; difference = 3.89, 95% CI: 4.51–7.32).

## 4. Discussion

The aims of the current study were to estimate the prevalence of psychological distress in the Australian community during the COVID-19 pandemic and to assess the psychometric properties of a new single-item measure of mental distress. Our results indicate there was a significant increase in levels of likely serious mental illness in Australia, from 6.3% pre-pandemic to 17.7% in early July 2020. The survey was in the field after the restrictions associated with the first wave of COVID-19 had lifted, and just as the first signs of community transmission that would become the second wave in Australia were being observed in the state of Victoria. However, we find psychological distress was not elevated in respondents from Victoria compared to the rest of Australia.

The current results are consistent with other Australian research using the K6, though based on a different methodology conducted at a similar time (17.1%) [8], but much greater than an Australian longitudinal cohort study that conducted fieldwork in April 2020 (10.6%) [4]. Other Australian studies using different measures of mental illness (e.g., PHQ) generated estimates of depression higher than our estimate of likely serious mental illness, specifically 20.3% [6] and 27.6% [7].

Our analysis of the change in psychological distress in different subpopulations identifies three groups in which the increase was most pronounced. Consistent with other studies [4,43], we found that young people showed a dramatic (almost four-fold) increase in distress. For example, in their longitudinal analysis based on the UK Understanding Society study, Pierce et al. [43] showed that the increase in psychological distress was greatest for younger people, and with no effect evident for those aged 55 years and older.

Our finding that distress increased more for men than women is contrary to most other research during COVID-19 [43]. Consistent with official employment statistics, we observed a large increase in the proportion of the population who were unemployed and, given the poorer mental health of those who are unemployed compared to those in employment, this contributed to the increase in psychological distress in the population. However, the major driver of the decline in the population’s mental health was the increase in distress among those who remained in employment. This is consistent with findings for, for example, the UK [43,44]. The COVID-19 pandemic eliminated the mental health advantage that people in employment usually have over those who are unemployed. Finally, it is worth noting that the results do not show a universal decline in mental health for all population groups. Older adults, those who were unemployed, and those not actively participating in the workforce showed no significant change in distress from pre-COVID levels. Again, these results are supported by that of Pierce et al. [43]. However, it is worth noting that Pierce et al. [43] reported that being retired in the years prior to the pandemic was associated with a substantial increase in psychological distress in fully adjusted models; a sub-group that we were not able to directly examine in our analyses. Overall, the specificity of the results increases confidence in the robustness of the current results.

There is a strong correlation (rho = 0.81) between scores on the single-item TTPN measure and the K6 scale, and moderate to high correspondence between these two measures of distress according to the ROC curve analysis with an AUC of 0.93 [42]. The optimal cut-point on the single item distress measure (between *some of the time* and *all of the time* response categories) results in the same classification decision as the recommended threshold of 18 on the K6 (distressed vs. not distressed) for 81% of respondents, and shows moderate sensitivity (0.71) and very high specificity (0.93). The Positive and Negative Predictive Values were broadly consistently with the sensitivity and specificity measures, respectively. These figures need to be interpreted in the context of the relatively low prevalence of distress in the population (17.6% in the current study according to the K6) which will deflate PPV and inflate NPV relative to samples in which the prevalence of distress is higher. Overall, the TTPN distress item is a useful addition to the researcher’s toolkit, providing an adequate measure of mental distress that can be used in circumstances where time is highly constrained but a measure of distress is needed. As described earlier, this may be the case when seeking to minimise respondent burden, when the costs of including a longer distress scale are prohibitive, or where the mode of administration preclude a longer questionnaire [18,19,26]. However, the TTPN distress item did over-estimate levels of distress among older respondent relative to the K6.

There are several limitations of this study that need to be acknowledged. First, the comparison to pre-pandemic distress is based on data collected using a different methodology and recruitment methods. However, our estimates are consistent with other Australian studies [8] and lower than that observed in similar international studies [45]. Second, the distress items in the TTPN survey use a 1-week reference period, asking respondents to report on distress experienced in the past week rather than the standard 4-week period usually used with the K6 (including the HILDA Survey). This reflects the fact that the TTPN survey was conducted weekly to monitor distress and other experiences in the rapidly changing COVID-19 environment. As mentioned, the developers of the K6 stated that changing the timeframe from 4 weeks to 1 week will not change scale interpretation [38]. Previous direct comparison of distress items using different reference periods (7 days and 30 days) showed items were rated similarly irrespective of the timeframe, although items referencing longer time periods had higher levels of endorsement [46]. If so, the differences between our pre-COVID measures of psychological distress based on the standard 4-week K6 and the 7-day TTPN item may underestimate the impact of the pandemic on levels of population distress. Third, our study evaluates the accuracy of the TTPN item against the K6, and does not involve validation against a clinical assessment or other gold-standard measures of mental disorders such as structured diagnostic instruments or symptom checklists [11,39]. Future research should examine the validity of the TTPN item with such a criterion. However, the main purpose of the new single-item distress measure was for within-sample comparison to evaluate trajectories of distress in Australia throughout the COVID-19 pandemic, while minimising respondent burden. The ability to capitalise on the wealth of representative data collected using the K6/10 as a benchmark for the TTPN item is a secondary, but nonetheless important, objective.

## 5. Conclusions

This study adds to the research literature demonstrating how the population’s mental health worsened during the COVID-19 pandemic. The increase was not uniform across all groups in society, and was most pronounced among men, young Australians, and those in employment. We also found that the single TTPN distress item provided a reasonable approximation of the estimate of distress generated by the K6. This analysis, therefore, provides the basis for future research drawing on the existing waves of the TTPN survey to examine trajectories and correlates of distress in Australia during the pandemic.

## Figures and Tables

**Figure 1 ijerph-19-00558-f001:**
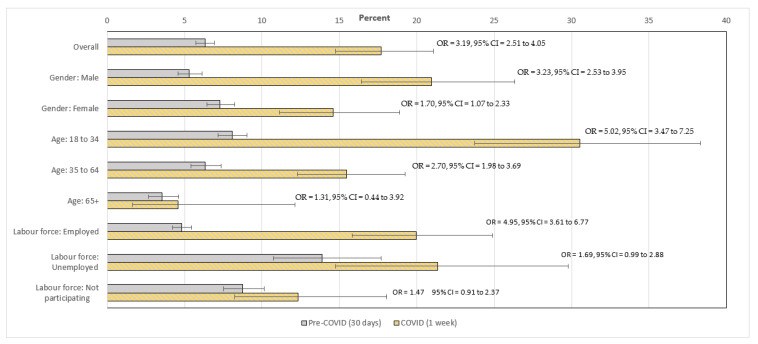
Weighted prevalence estimates of psychological distress (and standard errors) from the K6, comparing the 2019 wave of the HILDA Survey and the TTPN survey (wave 13), with odds ratios (and 95% confidence intervals) within each subgroup. The pre-COVID measure is the reference category for all analyses.

**Figure 2 ijerph-19-00558-f002:**
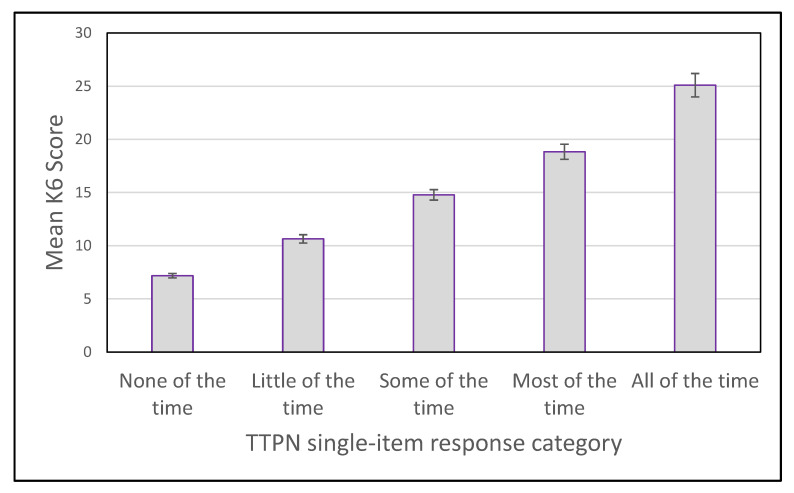
Mean K-6 score for each TTPN item category, with 95% confidence interval.

**Figure 3 ijerph-19-00558-f003:**
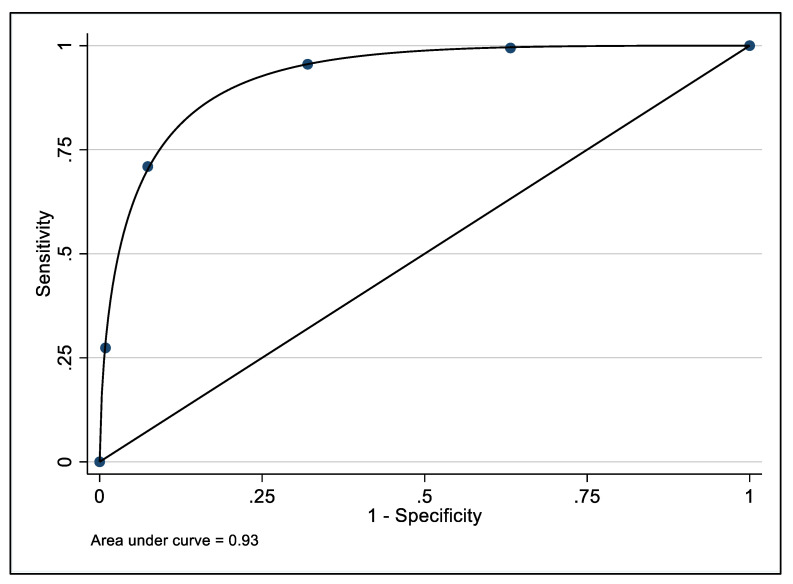
Receiver operating characteristic curve for the TTPN single item measure of mental distress against established K6 measure of psychological distress.

**Table 1 ijerph-19-00558-t001:** Unweighted characteristics of analysis sample from the 2019 wave of the Household Income and Labour Dynamics in Australia (HILDA) Survey and wave 13 of the Taking the Pulse of the Nation (TTPN) survey.

	Pre-COVID		COVID		
	HILDA Survey 2019		TTPN 2020 Wave 13		
Characteristic	Number	Percent	Number	Percent	Test of Unweighted Difference
Analysis sample	15,226	100	1150	100	
Gender					
Men	7136	46.9	572	49.7	*χ*^2^(1) = 3.54, *p* = 0.06
Women	8090	53.1	578	50.2	
Age group					
18–24 years	1721	4.3	108	9.4	*χ*^2^(6) = 39.75, *p* < 0.001
25–34 years	3053	20.1	191	16.6	
35–44 years	2360	15.5	204	17.7	
45–54 years	2384	15.7	213	18.5	
55–64 years	2477	16.3	200	17.4	
65–74 years	1910	12.5	174	15.1	
75 years +	1321	8.7	60	5.2	
Location					
Metropolitan	9630	63.7	677	58.9	*χ*^2^(1) = 10.84, *p* = 0.001
Rural	5481	36.3	473	41.1	
Labour force status					
Employed	9819	64.5	631	54.9	*χ*^2^(2) = 188.3, *p* < 0.001
Unemployed	520	3.4	139	12.1	
Not in the labour force	4887	32.1	380	33.0	
TTPN Mental distress					
No	-		957	83.2	
Yes			193	16.8	
K6 Mental distress					
No	14,368	93.7	974	84.7	*χ*^2^(1) = 125.29, *p* < 0.001
Yes	958	6.3	176	15.3	

**Table 2 ijerph-19-00558-t002:** Measures of correspondence between the binary measure of distress from the K6 and the TTPN distress item.

Cut-Point		Sensitivity	Specificity	Accurately Classified	Positive Predictive Validity	Negative Predictive Validity
≥1	(none of the time)	100.0	0	15.2	15.4	0
		–	-	(13.3–17.6)	(13.3–17.6)	-
≥2	(a little of the time)	99.4	37.2	46.8	22.3	99.7
		(96.9–99.9)	(34.1–40.4)	(43.9–49.7)	(19.5–25.4)	(98.5–99.9)
≥3	(some of the time)	95.5	68.7	72.8	35.6	98.8
		(91.3–98.0)	(65.7–71.6)	(70.1–75.3)	(31.3–40.1)	(97.7–99.5)
**≥4**	**(most of the time)**	**71.3**	**93.0**	**89.6**	**64.8**	**94.7**
		**(64.1–77.9)**	**(91.2–94.5)**	**(87.7–91.3)**	**(57.7–71.5)**	**(93.1–96.0)**
≥5	(all of the time)	27.5	99.3	88.3	87.5	88.3
		(21.1–34.7)	(98.5–99.7)	(86.3–90.1)	(75.9–94.8)	(86.2–90.1)

Note: Bold indicates recommended cut-point on the TTPN item. 95% confidence intervals are reported in brackets.

**Table 3 ijerph-19-00558-t003:** Random-effects logistic regression results comparing the two binary mental distress measures derived from the Table 2. including baseline model and series of models adding the interaction between measure and each socio-demographic characteristic.

	Base Model	Model 1 (Gender Interaction)	Model 2 (Age Interaction)	Model 3 (Area Interaction)	Model 4 (Labour Force Status Interaction)
Measure (ref = K6)	1.32	**1.78**	0.77	1.23	1.06
	(0.92–1.87)	**(1.06–3.00)**	(0.43–1.37)	(0.79–1.93)	(0.67—1.67)
Female (ref = Male)	**2.23**	**3.05**	**2.26**	**2.23**	**2.24**
	**(1.17–4.24)**	**(1.43–6.49)**	**(1.18–4.33)**	**(1.17–4.25**)	**(1.17–4.27)**
Age (ref = 18–34 years)					
Age 35–64	**0.20**	**0.20**	**0.14**	**0.20**	**0.20**
	**(0.10–0.42)**	**(0.10–0.42)**	**(0.06–0.32)**	**(0.10–0.42)**	**(0.10–0.42)**
Age 65+	**0.02**	**0.01**	**0.00**	**0.01**	**0.01**
	**(0.01–0.05)**	**(0.01–0.05)**	**(0.00–0.02)**	**(0.01–0.05)**	**(0.01–0.05)**
Location (ref = metropolitan)					
Rural	0.59	0.59	0.58	0.54	0.59
	(0.31–1.14)	(0.30–1.14)	(0.30–1.14)	(0.25–1.16)	(0.30–1.14)
Labour force status (ref = employed)					
Unemployed	2.08	2.09	2.10	2.08	1.46
	(0.82–5.28)	(0.82–5.35)	(0.81–5.43)	(0.82–5.29)	(0.49–4.36)
NILF	2.02	2.02	2.03	2.02	1.58
	(0.86–4.71)	(0.86–4.75)	(0.86–4.81)	(0.86–4.72)	(0.60–4.16)
Female × Measure		0.56			
		(0.27–1.14)			
Age 35–64 × Measure			2.07		
			(0.97–4.45)		
Age 75+ × Measure			**5.67**		
			**(1.39–23.12)**		
Location (Rural) × Measure				1.19	
				(0.58–2.47)	
Unemployed × Measure					1.95
					(0.71–5.36)
NILF × Measure					1.59
					(0.68–3.72)
Log-likelihood $\chi_{\left(1\right)}^{2}$		2.68	8.15	0.25	2.41
Prob > $\chi_{\left(1\right)}^{2}$		0.10	0.02	0.62	0.30

Note: Measure is the single-item TTPN mental distress measure. NILF denotes ‘Not in the Labour Force’. Coefficients are odds ratios with 95% confidence intervals in square brackets. Log-likelihood test reflects test statistic of estimated model with main-effects only model. Bold indicates coefficient is significant at *p* < 0.05.

## Data Availability

The raw data are not publicly available, but can be obtained via application to the Melbourne Institute: Applied Economic & Social Research.

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
