# Peer review of "Evaluating How Mental Health Changed in Australia through the COVID-19 Pandemic: Findings from the ‘Taking the Pulse of the Nation’ (TTPN) Survey"

_ijerph, 2022, doi:10.3390/ijerph19010558_

Round 1

Reviewer 1 Report

Abstract

  • Results section need some stats on validation. Need to include some inferential results obtained from the fitted models (e.g. ORs with 95% CI)) along with the descriptive stats (percentage change).

Introduction

  • It is mentioned that 43 separate waves of data collection occurred. What type of study design was used? E.g. cross-sectional or time series. How the time factor was considered in the evaluation? Although, in the methods section it is mentioned that cross-sectional study design was used.

Materials and methods:

  • How the weights generated for each wave were considered in the fitted models? Models need to adjust for weights. Not clear how it was done. Need detailed information.
  • It is not clear how the missing values were deleted e.g. list wise? Although, percentage of missing were less than 5% but there should be some references to validate the selected method.  
  • Under measures, coding of the outcome and input variables are missing. Before introducing statistical analysis, it is important to describe the outcome and input variables. Explain how and why binary logistic regression models are appropriate etc and use adjusted models by controlling for covariates. Also, it is not mentioned why only selected covariates are included in the study, some references would be useful.
  • Only Cronbach Alpha was used for the internal consistency measure of K6 scale. There are other supporting reliability stats should be presented along with Cronbach Alpha e.g. inter-item correlations etc. Also, 95% CI for the Cronbach Alpha should be presented along with Alpha.
  • Need some explanation how the logistic regression models were adjusted for weights and how the outcomes were controlled for covariates.
  • Before fitting the logistic regression models, bivariate analysis should have been conducted to assess the association between outcome and covariates along with their distributions.  
  • Why random-effects models were used and how the weights were adjusted in these models?
  • What the benefit of interaction effects considered in the models given that none of them was significant?

Results

  • First paragraph of the results section needs some work. Looks like bivariate analysis was partly conducted but presentation of results is not clear. Results of the bivariate analyse presented partly in lines 272-274. However, the analysis should not be for interaction test, it should be for association test.
  • Table 1 in not meaningful, sample size and percentage (n(%) can be presented in one column and results of the bivariate analysis (Chi-square and p-values) should be included along with these two columns.
  • Results obtained from bivariate analysis should be unweighed not weighted.
  • All bivariate analysis should be conducted before fitting the regression models.

  • Forest plots of the ORs presented in Figure 1 need some clarification. Not sure how weights were adjusted in the fitted models. For the adjusted model OR should be denoted as Adjusted OR and for the unadjusted modes these should be presented as Unadjusted OR. These can be done using notes underneath the figure. Also, reference category for the binary regression model should be indicated.
  • Tables 2-3 need some work, column titles required name of the stats presented in the table e.g. stats (95% CI).
  • Table 3: interactions were not significant. What are the benefits of including these in the models?
  • Interpretation of the ORs are required, only presenting the OR values not meaningful to the reader. For example, in line 233 it is written ‘with an Odds Ratio of 3.19’. This should be presented in plain language e.g. x times higher/lower….. This should be revised throughout the whole results section.

Limitation:

  • Limitation about missing value and cross-section studies should be included. Any limitation with response/non-response bias?

Author Response

Reviewer 1

  • Abstract: Results section need some stats on validation. Need to include some inferential results obtained from the fitted models (e.g. ORs with 95% CI)) along with the descriptive stats (percentage change).

Response: This has been added to the introduction.

  • Introduction: It is mentioned that 43 separate waves of data collection occurred. What type of study design was used? E.g. cross-sectional or time series. How the time factor was considered in the evaluation? Although, in the methods section it is mentioned that cross-sectional study design was used.

Response: Our point here was simply to say that 43 waves of the TTPN survey have been conducted. As described in the data section later in Section 2, however, we explicitly state that in this study we only use wave 13 of the TTPN, as it is the only wave that included both the single-item distress measure and the K6 measure, and is thus the only TTPN wave that allows us to answer our research question of agreement between the two mental health measures. Where we mention the 43 waves in the introduction, we have now also added text to refer the reader to Section 2.

  • How the weights generated for each wave were considered in the fitted models? Models need to adjust for weights. Not clear how it was done. Need detailed information.

Response: As stated in Section 2.1, the TTPN sample is stratified by age, gender, and population to be representative of the Australian population. This survey design is accounted for in the analyses that we do. The analysis is based on data from a single wave of the TTPN survey that included both the K6 and the TTPN distress item. Probability weights applied to strata based on age, sex and location were designed to reflect the Australia population and were included using the Stata svy: command. Also see Section 2.5.

  • It is not clear how the missing values were deleted e.g. list wise? Although, percentage of missing were less than 5% but there should be some references to validate the selected method.  

Response: In lines 150-157 we describe that it is due to listwise deletion/complete case analysis – and the discussion directly after that suggests little bias in missingness.

  • Under measures, coding of the outcome and input variables are missing. Before introducing statistical analysis, it is important to describe the outcome and input variables. Explain how and why binary logistic regression models are appropriate etc and use adjusted models by controlling for covariates. Also, it is not mentioned why only selected covariates are included in the study, some references would be useful.

Response: In the original version of the paper, we explicitly stated that coding and categories of all variables used, and we do so again in the revised version. An explanation for the rationale of the logistic regression models is provided in the ‘statistical analysis section’. We also note on line 167 and 179 that we consider a binary measure of distress based on the established cut-point on the K6, and similar on the TTPN item. In line 186 we have added text about the choice of controls; this is because they are the only ones in the data that are expected to likely be associated with distress (given that the TTPN survey is relatively brief, only a limited number of questions can be asked and hence only limited control variables are available. 

  • Only Cronbach Alpha was used for the internal consistency measure of K6 scale. There are other supporting reliability stats should be presented along with Cronbach Alpha e.g. inter-item correlations etc. Also, 95% CI for the Cronbach Alpha should be presented along with Alpha.

Response: We have now added bootstrap 95% CI’s and added item-rest correlations (average and range) under Section 2.3.

  • Need some explanation how the logistic regression models were adjusted for weights and how the outcomes were controlled for covariates.

Response: Please also refer to the response to a previous comment where we discuss how the weights were used. In the methods section, we also explain which covariates are used.

  • Before fitting the logistic regression models, bivariate analysis should have been conducted to assess the association between outcome and covariates along with their distributions.

Response: In the revised version, we have added the bivariate statistics to Table 1.

  • Why random-effects models were used and how the weights were adjusted in these models?

Response: As described in more detail on line 234, this was done to conduct a within-person comparison of prevalence based on the two different approaches, and whether they differ by key characteristics. This analysis was unweighted (see line 222).

  • What the benefit of interaction effects considered in the models given that none of them was significant?

Response: Please also refer to the response to a similar question below. These interactions tested not only whether the two different ways of measuring psychological distress (K6 and TTPN) produced similar overall measures, but whether they were consistent across different socio-demographic groups. The analysis did find one significant interaction involving age. Thus, as we now note more clearly on line 420, the TTPN distress item did over-estimate levels of distress in older respondents relative to the K6. The fact that only one of the interaction coefficients was significant is important for our purposes, as it demonstrates that the distress measures are similar across different sub-groups.

  • First paragraph of the results section needs some work. Looks like bivariate analysis was partly conducted but presentation of results is not clear. Results of the bivariate analyse presented partly in lines 272-274. However, the analysis should not be for interaction test, it should be for association test. 

Response: We included unweighted bivariate comparisons in Table 1 and the text – almost all comparisons were significant. And then we also describe in text the results of weighted analysis – showing characteristics we expect to be stable over time do not differ when weighted to population parameters, but distress and labour force status do change from pre-COVID to COVID. Also refer to response to next comment.

  • Table 1 in not meaningful, sample size and percentage (n(%) can be presented in one column and results of the bivariate analysis (Chi-square and p-values) should be included along with these two columns.

Response: From a presentation point of view, the table looked good when keeping the sample size and percentage columns as is. As suggested, we have included the bivariate analysis results in an additional column.

  • Results obtained from bivariate analysis should be unweighed not weighted.

Response: Noted. We have reported the unweighted results in the table, and also mentioned the weighted results in the text. It is still important to demonstrate that the weights remove unexpected differences between the two surveys used.

  • All bivariate analysis should be conducted before fitting the regression models.

Response: These analyses were conducted and are reported in Table 1 as well as the corresponding text in Section 3.1.

  • Forest plots of the ORs presented in Figure 1 need some clarification. Not sure how weights were adjusted in the fitted models. For the adjusted model OR should be denoted as Adjusted OR and for the unadjusted modes these should be presented as Unadjusted OR. These can be done using notes underneath the figure. Also, reference category for the binary regression model should be indicated.

Response: The OR’s in Figure 1 are unadjusted – and the ORs refer to the stratified analysis within each characteristic. We note this in the table title. We also state that the pre-COVID measure is the reference category for all analyses.

  • Tables 2-3 need some work, column titles required name of the stats presented in the table e.g. stats (95% CI).

Response: We have added text to the notes in both tables to make it clearer what the statistics are, e.g. that 95% CI’s are in brackets and also what the bold font represents in these tables. In terms of column headings for Table 3, the numbered headings simply refer to the different versions of the same model. Nevertheless, in the revised version we have added more informative headings to the columns.

  • Table 3: interactions were not significant. What are the benefits of including these in the models? 

Response: These tested not only whether the two different ways of measuring psychological distress (K6 and TTPN) produced similar overall measures, but whether they were consistent across different socio-demographic groups. The fact that the interaction coefficients are not significant is important for our purposes, as it demonstrates that the distress measures are similar across the different sub-groups.

  • Interpretation of the ORs are required, only presenting the OR values not meaningful to the reader. For example, in line 233 it is written ‘with an Odds Ratio of 3.19’. This should be presented in plain language e.g. x times higher/lower….. This should be revised throughout the whole results section.

Response: We have added text to the relevant part to formally interpret what these OR’s mean. Given this initial interpretation, it should be straightforward for readers to interpret remaining OR’s that are mentioned in the section, of which there are not that many.

  • Limitation about missing value and cross-section studies should be included. Any limitation with response/non-response bias?

Response: Because we primarily look at changes in distress during the COVID period relative to a year before, and because we then also only look at agreement between the single-item distress measure and the K6, we did not deem it fitting to list limitations of cross-section studies here, as these likely are not as applicable in this case. In our sample, only about 4% of the original sample had missing values on the main variables. As reported in Section 2.2., our analysis shows that those excluded with missing data were similar to those without missing data on all characteristics (gender, labour force status, residential location (metropolitan/rural) except for age; younger respondents were less likely to complete all items. Overall, therefore, we are not overly concerned about the extent and nature of missing values and non-response in the sample.

Reviewer 2 Report

  • Thanks for your effort working on this paper!
  • This study is significant in that it accumulated empirical evidence for research in related fields by confirming the impact of COVID-19 Pandemic on the psychological distress of Australians.
  • It is also meaningful in that it suggests the applicability of the new single-item measure of distress by revealing that it has a significant correlation with K6.
  • The research method is not unreasonable, the discussion is well described according to the study results, and it guides the direction of the follow-up research by clarifying the limitations of the research in detail.
  • Please correct the first letter of ‘survey’ in the title to uppercase.
  • Additional comments: This study is meaningful in that it improved the understanding of related fields by comparatively analyzing the prevalence of psychological distress        changed in Australia during the pandemic through comparison with pre-covid. It is also meaningful in that it introduced a new single-item measure of        distress as a simple and cost-effective tool to measure the psychological distress of the subject.
  • However, in the discussion, please discuss in detail the evidence for the increase in psychological distress among young people and men by comparing them with previous studies. Also, please add a discussion of the findings showing no difference in psychological distress compared to from pre-COVID levels for older people who are not employed and not actively participating in the workforce.
  •  

Author Response

Reviewer 2

  • Please correct the first letter of ‘survey’ in the title to uppercase.

Response: The change has been made.

  • In the discussion, please discuss in detail the evidence for the increase in psychological distress among young people and men by comparing them with previous studies. Also, please add a discussion of the findings showing no difference in psychological distress compared to from pre-COVID levels for older people who are not employed and not actively participating in the workforce.

Response: In the discussion section we have now added relevant references to the statements about increases in distress for the different sub-groups.

Reviewer 3 Report

The investigators study the prevalence of high psychological distress in the Australian population, and among key subpopulations, during the COVID-19 pandemic using the K6. Also, compare the results with pre-COVID population estimates. At the same time, they assess the psychometric properties and the accuracy of a single-item measure of mental distress in comparison to the K6. The data emanate from the Melbourne Institute’s Taking the Pulse of the Nation (TTPN) 118 survey, a repeated cross-sectional survey conducted in Australia since April 2020. As expected the results showed that there was a significant increase in the levels of probable serious mental illness in Australia due to the Covid-19 pandemic.

The study is interesting, although it has some limitations in that the data collected and the recruitment carried out in the pre-pandemic period differs from that used in the present investigation.

Author Response

Reviewer 3

  • The study is interesting, although it has some limitations in that the data collected and the recruitment carried out in the pre-pandemic period differs from that used in the present investigation.

Response: We acknowledged in the limitations section of the paper that this is indeed the case. Unfortunately, there is not much that can be done about that. However, the fact that the TTPN distress item and K6 are so highly comparable (as we show in our results) suggests that using the pre-COVID HILDA data (for which we use the K6) as comparison with the TTPN should be very reliable.

Round 2

Reviewer 1 Report

Revised version looks good.